# IMPROVED STEIN VARIATIONAL GRADIENT DESCENT WITH IMPORTANCE WEIGHTS

## ABSTRACT

Stein Variational Gradient Descent (SVGD) is a popular sampling algorithm used in various machine learning tasks. It is well known that SVGD arises from a discretization of the kernelized gradient flow of the Kullback-Leibler divergence $D_{\mathrm{KL}}\left(\cdot \mid \pi\right)$, where $\pi$ is the target distribution. In this work, we propose to enhance SVGD via the introduction of *importance weights*, which leads to a new method for which we coin the name $\beta$-SVGD. In the continuous time and infinite particles regime, the time for this flow to converge to the equilibrium distribution $\pi$, quantified by the Stein Fisher information, depends on $\rho_0$ and $\pi$ very weakly. This is very different from the kernelized gradient flow of Kullback-Leibler divergence, whose time complexity depends on $D_{\mathrm{KL}}\left(\rho_0 \mid \pi\right)$. Under certain assumptions, we provide a descent lemma for the population limit $\beta$-SVGD, which covers the descent lemma for the population limit SVGD when $\beta \to 0$. We also illustrate the advantages of $\beta$-SVGD over SVGD by simple experiments.

## 1 INTRODUCTION

The main technical task of Bayesian inference is to estimate integration with respect to the posterior distribution

$$\pi(x) \propto e^{-V(x)},$$

where $V : \mathbb{R}^d \to \mathbb{R}$ is a potential. In practice, this is often reduced to sampling points from the distribution $\pi$. Typical methods that employ this strategy include algorithms based on Markov Chain Monte Carlo (MCMC), such as Hamiltonian Monte Carlo (Neal, 2011), also known as Hybrid Monte Carlo (HMC) (Duane et al., 1987; Betancourt, 2017), and algorithms based on Langevin dynamics (Dalalyan & Karagulyan, 2019; Durmus & Moulines, 2017; Cheng et al., 2018).

One the other hand, Stein Variational Gradient Descent (SVGD)—a different strategy suggested by Liu & Wang (2016)—is based on an interacting particle system. In the population limit, the interacting particle system can be seen as the kernelized negative gradient flow of the Kullback-Leibler divergence

$$D_{\mathrm{KL}}\left(\rho \mid \pi\right) := \int \log\left(\tfrac{\rho}{\pi}\right)(x)\, d\rho(x); \tag{1}$$

see (Liu, 2017; Duncan et al., 2019). SVGD has already been widely used in a variety of machine learning settings, including variational auto-encoders (Pu et al., 2017), reinforcement learning (Liu et al., 2017), sequential decision making (Zhang et al., 2018; 2019), generative adversarial networks (Tao et al., 2019) and federated learning (Kassab & Simeone, 2022). However, current theoretical understanding of SVGD is limited to its infinite particle version (Liu, 2017; Korba et al., 2020; Salim et al., 2021; Sun et al., 2022), and the theory on finite particle SVGD is far from satisfactory.

Since SVGD is built on a discretization of the kernelized negative gradient flow of (1), we can learn about its sampling potential by studying this flow. In fact, a simple calculation (for example, see Korba et al. (2020)) reveals that

$$\min_{0 \le s \le t} I_{Stein}\left(\rho_s \mid \pi\right) \le \frac{D_{\mathrm{KL}}(\rho_0 \mid \pi)}{t}, \tag{2}$$

where $I_{Stein}\left(\rho_s \mid \pi\right)$ is the Stein Fisher information (see Definition 2) of $\rho_s$ relative to $\pi$, which is typically used to quantify how close to $\pi$ are the probability distributions $\left(\rho_s\right)_{s=0}^t$ generated along

this flow. In particular, if our goal is to guarantee $\min_{0 \le s \le t} I_{Stein}\left(\rho_s \mid \pi\right) \le \varepsilon$, result (2) says that we need to take

$$t \ge \frac{D_{\mathrm{KL}}(\rho_0 \mid \pi)}{\varepsilon}.$$

Unfortunately, and this is the key motivation for our work, the quantity the initial KL divergence $D_{\mathrm{KL}}\left(\rho_0 \mid \pi\right)$ can be very large. Indeed, it can be proportional to the underlying dimension, which is highly problematic in high dimensional regimes. Salim et al. (2021) and Sun et al. (2022) have recently derived an iteration complexity bound for the infinite particle SVGD method. However, similarly to the time complexity of the continuous flow, their bound depends on $D_{\mathrm{KL}}\left(\rho_0 \mid \pi\right)$.

## 1.1 SUMMARY OF CONTRIBUTIONS

In this paper, we design a family of continuous time flows—which we call $\beta$-SVGD flow—by combining *importance weights* with the kernelized gradient flow of the KL-divergence. Surprisingly, we prove that the time for this flow to converge to the equilibrium distribution $\pi$, that is $\min_{0 \le s \le t} I_{Stein}\left(\rho_s \mid \pi\right) \le \varepsilon$ with $(\rho_s)_{s=0}^t$ generated along $\beta$-SVGD flow, can be bounded by $-\frac{1}{\varepsilon \beta(\beta+1)}$ when $\beta \in (-1, 0)$. This indicates that the importance weights can potentially accelerate SVGD. Actually, we design $\beta$-SVGD method based on a discretization of the $\beta$-SVGD flow and provide a descent lemma for its population limit version. Some simple experiments in Appendix D verify our predictions.

We summarize our contributions in the following:

- **A new family of flows.** We construct a family of continuous time flows for which we coin the name $\beta$-SVGD flows. These flows do *not* arise from a time re-parameterization of the SVGD flow since their trajectories are different, nor can they be seen as the kernelized gradient flows of the Rényi divergence.

- **Convergence rates.** When $\beta \to 0$, this returns back to the kernelized gradient flow of the KL-divergence (SVGD flow); when $\beta \in (-1, 0)$, the convergence rate of $\beta$-SVGD flows is significantly improved than that of the SVGD flow in the case $D_{\mathrm{KL}}\left(\rho_0 \mid \pi\right)$ is large. Under a Stein Poincaré inequality, we derive an exponential convergence rate of 2-Rényi divergence along 1-SVGD flow. Stein Poincaré inequality is proved to be weaker than Stein log-Sobolev inequality, however like Stein log-Sobolev inequality, it is not clear to us when it does hold.

- **Algorithm.** We design $\beta$-SVGD algorithm based on a discretization of the $\beta$-SVGD flow and we derive a descent lemmas for the population limit $\beta$-SVGD.

- **Experiments.** Finally, we do some experiments to illustrate the advantages of $\beta$-SVGD with negative $\beta$. The simulation results on $\beta$-SVGD when $\beta$ changes from positive to negative corroborate our theory.

## 1.2 RELATED WORKS

The SVGD sampling technique was first presented in the fundamental work of Liu & Wang (2016). Since then, a number of SVGD variations have been put out. The following is a partial list: Newton version SVGD (Detommaso et al., 2018), stochastic SVGD (Gorham et al., 2020), mirrored SVGD (Shi et al., 2021), random-batch method SVGD (Li et al., 2020) and matrix kernel SVGD (Wang et al., 2019). The theoretical knowledge of SVGD is still constrained to population limit SVGD. The first work to demonstrate the convergence of SVGD in the population limit was by Liu (2017); Korba et al. (2020) then derived a similar descent lemma for the population limit SVGD using a different approach. However, their results relied on the path information and thus were not self-contained, to provide a clean analysis, Salim et al. (2021) assumed a Talagrand's $T_1$ inequality of the target distribution $\pi$ and gave the first iteration complexity analysis in terms of dimension $d$. Following the work of Salim et al. (2021); Sun et al. (2022) derived a descent lemma for the population limit SVGD under a non-smooth potential $V$.

In this paper, we consider a family of generalized divergences, Rényi divergence, and SVGD with importance weights. For these two themes, we name a few but non-exclusive related results. Wang et al. (2018) proposed to use the $f$-divergence instead of KL-divergence in the variational inference

problem, here $f$ is a convex function; Yu et al. (2020) also considered variational inference with $f$-divergence but with its dual form. Han & Liu (2017) considered combining importance sampling with SVGD, however the importance weights were only used to adjust the final sampled points but not in the iteration of SVGD as in this paper; Liu & Lee (2017) considered importance sampling, they designed a black-box scheme to calculate the importance weights (they called them Stein importance weights in their paper) of any set of points.

## 2 PRELIMINARIES

We assume the target distribution $\pi \propto e^{-V}$, and we have oracle to calculate the value of $e^{-V(x)}$ for all $x \in \mathbb{R}^d$.

### 2.1 NOTATION

Let $x = (x_1, \ldots, x_d)^\top, y = (y_1, \ldots, y_d)^\top \in \mathbb{R}^d$, denote $\langle x, y \rangle := \sum_{i=1}^d x_i y_i$ and $\|x\| := \sqrt{\langle x, x \rangle}$. For a square matrix $B \in \mathbb{R}^{d \times d}$, the operator norm and Frobenius norm of $B$ are defined respectively by $\|B\|_{op} := \sqrt{\varrho(B^\top B)}$ and $\|B\|_F := \sqrt{\sum_{i=1}^d \sum_{j=1}^d B_{i,j}^2}$, respectively, where $\varrho$ denotes the spectral radius. It is easy to verify that $\|B\|_{op} \leq \|B\|_F$. Let $\mathcal{P}_2(\mathbb{R}^d)$ denote the space of probability measures with finite second moment; that is, for any $\mu \in \mathcal{P}_2(\mathbb{R}^d)$ we have $\int \|x\|^2 \ d\mu(x) < +\infty$. The Wasserstein 2-distance between $\rho, \mu \in \mathcal{P}_2(\mathbb{R}^d)$ is defined by

$$W_2(\rho, \mu) := \inf_{\eta \in \Gamma(\rho, \pi)} \sqrt{\int \|x - y\|^2 \ d\eta(x, y)},$$

where $\Gamma(\rho, \mu)$ is the set of all joint distributions defined on $\mathbb{R}^d \times \mathbb{R}^d$ having $\rho$ and $\mu$ as marginals. The push-forward distribution of $\rho \in \mathcal{P}_2(\mathbb{R}^d)$ by a map $T : \mathbb{R}^d \to \mathbb{R}^d$, denoted by $T_{\#}\rho$, is defined as follows: for any measurable set $\Omega \in \mathbb{R}^d$, $T_{\#}\rho(\Omega) := \rho(T^{-1}(\Omega))$. By definition of the push-forward distribution, it is not hard to verify that the probability densities satisfy $T_{\#}\rho(T(x))|\det D T(x)| = \rho(x)$, where $D T$ is the Jacobian matrix of $T$. The reader can refer to Villani (2009) for more details.

### 2.2 RÉNYI DIVERGENCE

Next, we define the Rényi divergence which plays an important role in information theory and many other areas such as hypothesis testing (Morales González et al., 2013) and multiple source adaptation (Mansour et al., 2012).

**Definition 1 (Rényi divergence)** *For two probability distributions $\rho$ and $\mu$ on $\mathbb{R}^d$ and $\rho \ll \mu$, the Rényi divergence of positive order $\alpha$ is defined as*

$$D_\alpha(\rho \mid \mu) := \begin{cases} \frac{1}{\alpha - 1} \log \left( \int \left( \frac{\rho}{\mu} \right)^{\alpha - 1} (x) \ d\rho(x) \right) & 0 < \alpha < \infty, \ \alpha \neq 1 \\ \int \log \left( \frac{\rho}{\mu} \right) (x) \ d\rho(x) & \alpha = 1 \end{cases}. \tag{3}$$

*If $\rho$ is not absolutely continuous with respect to $\mu$, we set $D_\alpha(\rho \mid \mu) = \infty$. Further, we denote $D_{KL}(\rho \mid \mu) := D_1(\rho \mid \mu)$.*

Rényi divergence is non-negative, continuous and non-decreasing in terms of the parameter $\alpha$; specifically, we have $D_{KL}(\rho \mid \mu) = \lim_{\alpha \to 1} D_\alpha(\rho \mid \mu)$. More properties of Rényi divergence can be found in a comprehensive article by Van Erven & Harremos (2014). Besides Rényi divergence, there are other generalizations of the KL-divergence, e.g., admissible relative entropies (Arnold et al., 2001).

### 2.3 BACKGROUND ON SVGD

Stein Variational Gradient Descent (SVGD) is defined on a Reproducing Kernel Hilbert Space (RKHS) $\mathcal{H}_0$ with a non-negative definite reproducing kernel $k : \mathbb{R}^d \times \mathbb{R}^d \to \mathbb{R}^+$. The

key feature of this space is its reproducing property:

$$f(x) = \langle f(\cdot), k(x, \cdot) \rangle_{\mathcal{H}_0}, \qquad \forall f \in \mathcal{H}_0, \tag{4}$$

where $\langle \cdot, \cdot \rangle_{\mathcal{H}_0}$ is the inner product defined on $\mathcal{H}_0$. Let $\mathcal{H}$ be the $d$-fold Cartesian product of $\mathcal{H}_0$. That is, $f \in \mathcal{H}$ if and only if there exist $f_1, \cdots, f_d \in \mathcal{H}_0$ such that $f = (f_1, \ldots, f_d)^\top$. Naturally, the inner product on $\mathcal{H}$ is given by

$$\langle f, g \rangle_{\mathcal{H}} := \sum_{i=1}^d \langle f_i, g_i \rangle_{\mathcal{H}_0}, \qquad f = (f_1, \ldots, f_d)^\top \in \mathcal{H}, \qquad g = (g_1, \ldots, g_d)^\top \in \mathcal{H}. \tag{5}$$

For more details of RKHS, the readers can refer to Berlinet & Thomas-Agnan (2011).

It is well known (see for example Ambrosio et al. (2005)) that $\nabla \log \left( \frac{\rho}{\pi} \right)$ is the Wasserstein gradient of $D_{\mathrm{KL}} \left( \cdot \mid \pi \right)$ at $\rho \in \mathcal{P}_2(\mathbb{R}^d)$. Liu & Wang (2016) proposed a kernelized Wasserstein gradient of the KL-divergence, defined by

$$g_\rho(x) := \int k(x, y) \nabla \log \left( \frac{\rho}{\pi} \right)(y) \, d\rho(y) \in \mathcal{H}. \tag{6}$$

Integration by parts yields

$$g_\rho(x) = -\int \left[ \nabla \log \pi(y) k(x, y) + \nabla_y k(x, y) \right] \, d\rho(y). \tag{7}$$

Comparing the Wasserstein gradient $\nabla \log \left( \frac{\rho}{\pi} \right)$ with (7), we find that the latter can be easily approximated by

$$g_\rho(x) \approx \hat{g}_{\hat{\rho}} := -\frac{1}{N} \sum_{i=1}^N \left[ \nabla \log \pi(x_i) k(x, x_i) + \nabla_{x_i} k(x, x_i) \right], \tag{8}$$

with $\hat{\rho} = \frac{1}{N} \sum_{i=1}^N \delta_{x_i}$ and $(x_i)_{i=1}^N$ sampled from $\rho$. With the above notations, the SVGD update rule

$$x_i \leftarrow x_i + \frac{\gamma}{N} \sum_{j=1}^N \left[ \nabla \log \pi(x_j) k(x_i, x_j) + \nabla_{x_j} k(x_i, x_j) \right], \quad i = 1, \ldots, N, \tag{9}$$

where $\gamma$ is the step-size, can be presented in the compact form $\hat{\rho} \leftarrow (I - \gamma \hat{g}_{\hat{\rho}})_\# \hat{\rho}$. When we talk about the infinite particle SVGD, or population limit SVGD, we mean $\rho \leftarrow (I - \gamma g_\rho)_\# \rho$. The metric used in the study of SVGD is the Stein Fisher information or the Kernelized Stein Discrepancy (KSD).

**Definition 2 (Stein Fisher Information)** *Let $\rho \in \mathcal{P}_2(\mathbb{R}^d)$. The Stein Fisher Information of $\rho$ relative to $\pi$ is defined by*

$$I_{Stein}(\rho \mid \pi) := \iint k(x, y) \left\langle \nabla \log \left( \frac{\rho}{\pi} \right)(x), \nabla \log \left( \frac{\rho}{\pi} \right)(y) \right\rangle \, d\rho(x) \, d\rho(y). \tag{10}$$

A sufficient condition under which $\lim_{n \to \infty} I_{Stein}(\rho_n \mid \pi)$ implies $\rho_n \to \pi$ weakly can be found in Gorham & Mackey (2017), which requires: i) the kernel $k$ to be in the form $k(x, y) = \left( c^2 + \|x - y\|^2 \right)^\theta$ for some $c > 0$ and $\theta \in (-1, 0)$; ii) $\pi \propto e^{-V}$ to be distant dissipative; roughly speaking, this requires $V$ to be convex outside a compact set, see Gorham & Mackey (2017) for an accurate definition.

In the study of the kernelized Wasserstein gradient (7) and its corresponding continuity equation

$$\frac{\partial \rho_t}{\partial t} + \mathrm{div} \left( \rho_t g_{\rho_t} \right) = 0,$$

Duncan et al. (2019) introduced the following kernelized log-Sobolev inequality to prove the exponential convergence of $D_{\mathrm{KL}} \left( \rho_t \mid \pi \right)$ along the direction (7):

**Definition 3 (Stein log-Sobolev inequality)** *We say $\pi$ satisfies the Stein log-Sobolev inequality with constant $\lambda > 0$ if*

$$D_{\mathrm{KL}}(\rho \mid \pi) \leq \frac{1}{2\lambda} I_{Stein}(\rho \mid \pi). \tag{11}$$

While this inequality can guarantee an exponential convergence rate of $\rho_t$ to $\pi$, quantified by the KL-divergence, the condition for $\pi$ to satisfy the Stein log-Sobolev inequality is very restrictive. In fact, little is known about when (11) holds.

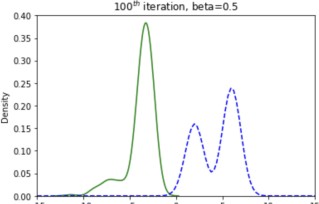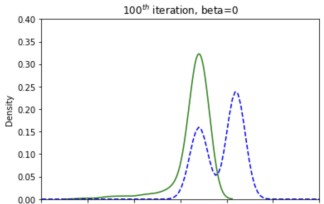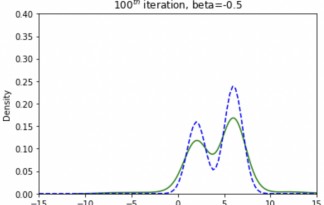

Figure 1: The performance of $\beta$-SVGD with three choices of $\beta$, but using the same step-size. The blue dashed line is the target distribution $\pi$: the Gaussian mixture $\frac{2}{5}\mathcal{N}(2,1) + \frac{3}{5}\mathcal{N}(6,1)$. The green solid line is the distribution generated by $\beta$-SVGD after 100 iterations; see Appendix D for more results and details.

## 3 CONTINUOUS TIME DYNAMICS OF THE $\beta$-SVGD FLOW

In this section, we mainly focus on the continuous time dynamics of the $\beta$-SVGD flow. Due to page limitation, we leave all of the proofs to Appendix B.

### 3.1 $\beta$-SVGD FLOW

In this paper, a *flow* refers to some time-dependent vector field $v_t : \mathbb{R}^d \to \mathbb{R}^d$. This time-dependent vector field will influence the mass distribution on $\mathbb{R}^d$ by the continuity equation or the equation of conservation of mass

$$\frac{\partial \rho_t}{\partial t} + \mathrm{div}\left(\rho_t v_t\right) = 0, \tag{12}$$

readers can refer to Ambrosio et al. (2005) for more details.

**Definition 4 ($\beta$-SVGD flow)** *Given a weight parameter $\beta \in (-1, +\infty)$, the $\beta$-SVGD flow is given by*

$$v_t^\beta(x) := -\left(\frac{\pi}{\rho_t}\right)^\beta(x) \int k(x,y)\nabla\log\left(\frac{\rho_t}{\pi}\right)(y)\,d\rho_t(y). \tag{13}$$

*Note that when $\beta = 0$, this is the negative kernelized Wasserstein gradient (6).*

Note that we can not treat $\beta$-SVGD flow as the kernelized Wasserstein gradient flow of the $(\beta + 1)$-Rényi divergence. However, they are closely related, and we can derive the following theorem.

**Theorem 1 (Main result)** *Along the $\beta$-SVGD flow (13), we have*[1]

$$\min_{t\in[0,T]} I_{Stein}\left(\rho_t \mid \pi\right) \leq \begin{cases} \frac{e^{\beta D_{\beta+1}(\rho_0|\pi)}}{T\beta(\beta+1)} & \beta > 0 \\ \frac{D_{\mathrm{KL}}(\rho_0|\pi)}{T} & \beta = 0 \\ -\frac{1}{T\beta(\beta+1)} & \beta \in (-1,0) \end{cases}. \tag{14}$$

Note the left hand side of (14) is the Stein Fisher information. When $\beta$ decreases from positive to negative, the right hand side of (14) changes dramatically; it appears to be independent of $\rho_0$ and $\pi$. If we do not know the Rényi divergence between $\rho_0$ and $\pi$, it seems the best convergence rate is obtained by setting $\beta = -\frac{1}{2}$, that is

$$\min_{t\in[0,T]} I_{Stein}\left(\rho_t \mid \pi\right) \leq \frac{4}{T}.$$

It is somewhat unexpected to observe that the time complexity is independent of $\rho_0$ and $\pi$, or to be more precise, that it relies only very weakly on $\rho_0$ and $\pi$ when $\beta \in (-1, 0)$. We wish to stress that this is *not* achieved by time re-parameterization. In the proof of Theorem 1, we can

---

[1]In fact, in the proof in Appendix B we know a stronger result. When $\beta \in (-1,0)$, the right hand side of (14) is only weakly dependent on $\rho_0$ and $\pi$ and should be $\frac{\left|e^{\beta D_{\beta+1}(\rho_0|\pi)} - e^{\beta D_{\beta+1}(\rho_T|\pi)}\right|}{T|\beta(\beta+1)|}$, which is less than $-\frac{1}{T\beta(\beta+1)}$.

see the term $(\pi/\rho_t)^\beta$ in $\beta$-SVGD flow (13) is utilized to cancel term $(\rho_t/\pi)^\beta$ in the Wasserstein gradient of $(\beta + 1)$-Rényi divergence. Actually, when $\beta \in (-1, 0)$, this term has an added advantage and can be seen as the acceleration factor in front of the kernelized Wasserstein gradient of KL-divergence. Specifically, the negative kernelized Wasserstein gradient of KL-divergence $v_t^0(x) := -\int k(x,y)\nabla \log(\frac{\rho_t}{\pi})(y)d\rho_t(y)$ is the vector field that compels $\rho_t$ to approach $\pi$, while $(\pi/\rho_t)^\beta(x)$ is big (roughly speaking this means $x$ is close to the mass concentration region of $\rho_t$ but away from the one of $\pi$), this factor will enhance the vector field at point $x$ and force the mass around $x$ move faster towards the mass concentration region of $\pi$; on the other hand, if $(\pi/\rho_t)^\beta(x)$ is small (this means $x$ is already near to the mass concentration region of $\pi$), this factor will weaken the vector field and make the mass surrounding $x$ remain within the mass concentration region of $\pi$. This is the intuitive justification for why, when $\beta \in (-1, 0)$, the time complexity for $\beta$-SVGD flow to diminish the Stein Fisher information only depends on $\rho_0$ and $\pi$ very weakly.

**Remark 1** *While it may seem reasonable to suspect that the time complexity of the $\beta$-SVGD flow with $\beta \leq -1$ will also depend on $\rho_0$ and $\pi$ very weakly, surprisingly, this is not true. In fact, we can prove that (see Appendix B)*

$$\min_{t \in [0,T]} I_{Stein}(\rho_t \mid \pi) \leq \frac{e^{(-\beta-1)D_{-\beta}(\pi|\rho_0)}}{|T\beta(\beta+1)|}.$$

*Letting $\beta \to -1$, we get $\min_{t\in[0,T]} I_{Stein}(\rho_t \mid \pi) \leq \frac{D_{\mathrm{KL}}(\pi|\rho_0)}{T}$. The regime when $\beta \leq -1$ is similar to the $\beta > 0$ regime in Theorem 1, which heavily depends on $\rho_0$ and $\pi$. Mathematically speaking, the weak dependence on $\rho_0$ and $\pi$ is caused by the concavity of the function $s^\alpha$ on $s \in \mathbb{R}^+$ when $\alpha = \beta + 1 \in (0, 1)$.*

## 3.2   1-SVGD FLOW AND THE STEIN POINCARÉ INEQUALITY

Functional $D_{\mathrm{KL}}(\cdot \mid \cdot)$ is non-symmetric; that is, $D_{\mathrm{KL}}(\cdot \mid \pi) \neq D_{\mathrm{KL}}(\pi \mid \cdot)$, and so is their Wasserstein gradient. The Wasserstein gradient of $D_{\mathrm{KL}}(\pi \mid \cdot)$ at distribution $\rho \in \mathcal{P}_2(\mathbb{R}^d)$ is $-\nabla\frac{\pi}{\rho}$ (see Appendix A), or, to put it another way, $\frac{\pi}{\rho}\nabla\log(\frac{\rho}{\pi})$, which may be regarded as the non-kernelized 1-SVGD flow (module a minus sigh) when compared to (13). To conclude, the 1-SVGD flow

$$v_t^1(x) := -\frac{\pi}{\rho_t}(x) \int k(x,y)\nabla\log\left(\frac{\rho_t}{\pi}\right)(y) \, d\rho_t(y), \tag{15}$$

is the negative kernelized Wasserstein gradient flow of $D_{\mathrm{KL}}(\pi \mid \cdot)$. Next, we will study the exponential convergence of 2-Rényi divergence along 1-SVGD flow under the Stein Poincaré inequality.

**Definition 5 (Stein Poincaré inequality)** *We say that $\pi$ satisfies the Stein Poincaré inequality with constant $\lambda > 0$ if*

$$\int |g|^2 \, d\pi \leq \frac{1}{\lambda} \iint k(x,y) \langle \nabla g(x), \nabla g(y) \rangle \, d\pi(x) \, d\pi(y), \tag{16}$$

*for any smooth $g$ with $\int g \, d\pi = 0$.*

While Duncan et al. (2019) also introduced the Stein Poincaré inequality, they presented it in a different form. Just as Poincaré inequality is a linearized log-Sobolev inequality (see for example (Bakry et al., 2014, Proposition 5.1.3)), Stein Poincaré inequality is also a linearized Stein log-Sobolev inequality (11). Although Stein Poincaré inequality is weaker than Stein log-Sobolev inequality, the condition for it to hold is quite restrictive, as in the case of Stein log-Sobolev inequality; see the discussion in (Duncan et al., 2019, Section 6).

**Lemma 1 (Stein log-Sobolev implies Stein Poincaré)** *If $\pi$ satisfies the Stein log-Sobolev inequality (11) with constant $\lambda > 0$, then it also satisfies the Stein Poincaré inequality with the same constant $\lambda$.*

While the proof of the above lemma is a routine task, for completeness we provide it in Appendix B. The following theorem is inspired by Cao et al. (2019), in which they proved the exponential convergence of Rényi divergence along Langevin dynamic under a strongly convex potential $V$. However, due to the structure of 1-SVGD flow, we can only prove the results for $\alpha$-Rényi divergence with $\alpha \in (0, 2]$.

**Theorem 2** *Suppose $\pi$ satisfies the Stein Poincaré inequality with constant $\lambda > 0$. Then the flow (15) satisfies*

$$\mathrm{D}_2\left(\rho_t \mid \pi\right) \leq C \cdot \mathrm{D}_2\left(\rho_0 \mid \pi\right) \cdot e^{-2\lambda t}, \tag{17}$$

*where $C = \frac{e^{\mathrm{D}_2(\rho_0|\pi)}-1}{\mathrm{D}_2(\rho_0|\pi)}$.*

Since $\mathrm{D}_{\alpha_1}\left(\rho \mid \pi\right) \leq \mathrm{D}_{\alpha_2}\left(\rho \mid \pi\right)$ for any $0 < \alpha_1 \leq \alpha_2 < \infty$, the exponential convergence of $\alpha$-Rényi divergence with $\alpha \in (0,2)$ can be easily deduced from (17).

**Corollary 1** *Suppose $\pi$ satisfies the Stein Poincaré inequality with constant $\lambda > 0$. Then the flow (15) satisfies*

$$\mathrm{D}_\alpha\left(\rho_t \mid \pi\right) \leq C \cdot \mathrm{D}_\alpha\left(\rho_0 \mid \pi\right) \cdot e^{-2\lambda t} \tag{18}$$

*for all $\alpha \in (0,2]$, where $C = \frac{e^{\mathrm{D}_2(\rho_0|\pi)}-1}{\mathrm{D}_\alpha(\rho_0|\pi)}$.*

## 4 THE $\beta$-SVGD ALGORITHM

The $\beta$-SVGD algorithm[2] proposed here is a sampling method suggested by the discretization of the $\beta$-SVGD flow (13). Our method reverts to the traditional SVGD algorithm when $\beta = 0$.

As in SVGD, the integral term in the $\beta$-SVGD flow (13) can be approximated by (8). However, when $\beta \neq 0$, we have to estimate the extra importance weight term $(\pi/\rho_t)^\beta$. We can use the kernel method (Silverman, 2018) to estimate $\rho_t$ given points sampled from $\rho_t$. The idea behind the kernel density estimation is simple. Assuming that Dirac $\delta(\cdot)$ can be weakly approximated by kernel $K(\cdot)$, that is for any bounded smooth function $f \in C_b^\infty(\mathbb{R}^d)$, we have

$$\lim_{h \to 0} \tfrac{1}{h^d} \int f(x) K\left(\tfrac{x}{h}\right) dx \to \int f(x)\delta(x)dx = f(0).$$

Then for $\rho$, a smooth probability density function on $\mathbb{R}^d$, we have

$$\rho(x) = \int \delta(x-y)\rho(y)\,dy \approx \tfrac{1}{h^d} \int K(\tfrac{x-y}{h})\rho(y)\,dy \approx \tfrac{1}{Nh^d} \sum_{i=1}^{N} K(\tfrac{x-y_i}{h}) =: \hat{\rho}(x),$$

with $(y_i)_{i=1}^N$ sampled from $\rho$. Usually, $K$ will be a radially symmetric unimodal probability density function, for example, the standard multivariate Gaussian

$$K_g(x) = \tfrac{1}{(2\Pi)^{\frac{d}{2}}} e^{-\frac{\|x\|^2}{2}}, \quad \Pi \text{ is the area of unit circle.}$$

While unfortunately we only know the value of $\pi(x)$ up to a normalizing constant, this constant is independent of $x$, allowing us to merge it into the step-size. One needs to keep in mind though that $(\pi/\rho_t)^\beta$ may explode from above. Therefore, in the implementation of $\beta$-SVGD (1), we must truncate this value from above by a relatively big number $M$.

**Remark 2** *Note that the performance of kernel density estimation largely depends on the sample size Parzen (1962); Devroye & Wagner (1979) and bandwidth $h$ Sheather (2004). Optimal bandwidth is difficult to obtain even with good bandwidth selection heuristics Scott & Sheather (1985). (Silverman, 2018, Section 4.3.1) showed that the approximately optimal bandwidth $h_{opt}$, in the sense of minimizing mean integrated square error (see the section in the book), should be of order $N^{-\frac{1}{d+4}}$. When $d = 1$, a lemma from Parzen (1962) (see also Lemma 2 in Appendix C) suggests $h \sim N^{-\frac{1}{2}}$.*

### 4.1 NON-ASYMPTOTIC ANALYSIS FOR $\beta$-SVGD

In this section, we study the convergence of the population limit $\beta$-SVGD. Specifically, we establish a descent lemma for it. The derivation of the descent lemma is based on several assumptions.

The first assumption postulates $L$-smoothness of $V$; this is typically assumed in the study of optimization algorithms, Langevin algorithms and SVGD.

---

[2]For simplicity, we will often just call it $\beta$-SVGD; not to be confused with the $\beta$-SVGD flow.

---

**Algorithm 1** Beta Stein Variational Gradient Descent ($\beta$-SVGD)

---

1: **Input:** The potential function $V : \mathbb{R}^d \to \mathbb{R}$ of the target distribution $\pi \propto e^{-V}$, density estimation kernel $K(\cdot)$, the scaling parameter $h$, reproducing kernel $k(\cdot, \cdot)$, a set of initial particles $(x_i^0)_{i=1}^N$ and iteration number $n$.

2: **for** $l = 0, 1, \dots, n$ **do**

3:     Estimate density: $\rho_i^l = \frac{1}{Nh^d} \sum_{j=1}^N K\left(\frac{x_i^l - x_j^l}{h}\right), \quad i = 1, \dots, N$

4:     Calculate the weight with choice 1: $w_i^l = \left[\left(\frac{e^{-V(x_i^l)}}{\rho_i^l}\right)^\beta \wedge M_l\right] \Big/ \left[\sum_{j=1}^N \left(\frac{e^{-V(x_j^l)}}{\rho_j^l}\right)^\beta \wedge M_l\right];$

   choice 2: $w_i^l = \left[\left(\frac{e^{-V(x_i^l)}}{\rho_i^l}\right)^\beta \wedge M_l\right] \Big/ N, \quad i = 1, \dots, N, \quad M_l$ is the truncating number

5:     Update particles with step-size $\gamma_l$: $x_i^{l+1} \leftarrow x_i^l + \gamma_l w_i^l \sum_{j=1}^N \left[-k(x_i^l, x_j^l) \nabla_{x_j^l} V(x_j^l) + \nabla_{x_j^l} k(x_i^l, x_j^l)\right], \quad i = 1, \dots, N$

6: **end for**

7: **Return:** Particles $(x_i^{n+1})_{i=1}^N$.

---

**Assumption 1** (*L-smoothness*) *The potential function $V$ of the target distribution $\pi \propto e^{-V}$ is L-smooth; that is,*

$$\left\|\nabla^2 V\right\|_{op} \leq L.$$

Our second assumption postulates two bounds involving the reproducing kernel $k(\cdot, \cdot)$, and is also common when studying SVGD; see (Liu, 2017; Korba et al., 2020; Salim et al., 2021; Sun et al., 2022).

**Assumption 2** *Kernel $k$ is continuously differentiable and there exists $B > 0$ such that* $\|k(x, .)\|_{\mathcal{H}_0} \leq B$ *and*

$$\|\nabla_x k(x, .)\|_{\mathcal{H}}^2 = \sum_{i=1}^d \|\partial_{x_i} k(x, .)\|_{\mathcal{H}_0}^2 \leq B^2, \qquad \forall x \in \mathbb{R}^d.$$

By the reproducing property (4), this is equivalent to $k(x, x) \leq B^2$ and $\sum_{i=1}^d \partial_{x_i} \partial_{y_i} k(x, y)\,|_{y=x} \leq B^2$ for any $x \in \mathbb{R}^d$, and this is easily satisfied by kernel of the form $k(x, y) = f(x - y)$, where $f$ is some smooth function at point 0.

The third assumption was already used by Liu (2017); Korba et al. (2020), and was later replaced by Salim et al. (2021) it with a Talagrand inequality (Wasserstein distance can be upper bounded by KL-divergence) which depends on $\pi$ only. However, $\beta$-SVGD reduces the Rényi divergence instead of the KL-divergence. Since we do not have a comparable inequality for the Rényi divergence, we are forced to adopt the one from (Liu, 2017; Korba et al., 2020) here.

**Assumption 3** *There exists $C > 0$ such that $\sqrt{I_{Stein}(\rho_n \mid \pi)} \leq C$ for all $n = 0, 1, \dots, N$.*

In the proof of the descent lemma, the next two assumptions help us deal with the extra term $(\pi/\rho_n)^\beta$. Note that the fourth assumption is very weak. In fact, as long as $Z_n(x, y)\rho_n(x)\rho_n(y)$ is integrable on $\mathbb{R}^d \times \mathbb{R}^d$, then by the monotone convergence theorem, the truncating number $M_{\rho_n}(\delta)$ is always attainable since $(\rho_n/\pi)^\beta (x) (\pi/\rho_n)^\beta \wedge M$ is non-decreasing and converges point-wise to 1 as $M \to +\infty$.

**Assumption 4** *For any small $\delta > 0$, we can find $M_{\rho_n}(\delta) > 0$ such that*

$$\left| I_{Stein}(\rho_n \mid \pi) - \iint \left(\frac{\rho_n}{\pi}\right)^\beta (x) \left(\frac{\pi}{\rho_n}\right)^\beta \wedge M_{\rho_n}(\delta) Z_n(x, y)\, d\rho_n(x)\, d\rho_n(y) \right| \leq \delta, \qquad (19)$$

*where $Z_n(x, y) := k(x, y) \left\langle \nabla \log\left(\frac{\rho_n}{\pi}\right)(x), \nabla \log\left(\frac{\rho_n}{\pi}\right)(y) \right\rangle$.*

Our fifth and last assumption is of a technical nature, and helps us bound $\left\|\nabla_x (\pi/\rho_n)^\beta (x) \left(\int k(x, y) \nabla \log(\frac{\rho_n}{\pi})(y) d\rho_n(y)\right)^\top\right\|_F$. It is also relatively weak, and achievable for example when the potential function of $\rho_n$ does not fluctuate wildly.

**Assumption 5** $\left\| \nabla \left( \frac{\pi}{\rho_n} \right)^{\beta} \right\| \leq C_{\rho_n}(\delta)$ *in the region* $\left\{ x : \left( \frac{\pi}{\rho_n} \right)^{\beta}(x) \leq M_{\rho_n}(\delta) \right\}$.

Though Assumptions 3, 4 and 5 are relatively reasonable, as we stated, we do not know how to estimate constants $C$, $M_{\rho_n}(\delta)$ and $C_{\rho_n}(\delta)$ beforehand.

With all this preparation, we can now formulate our descent lemma for the population limit $\beta$-SVGD when $\beta \in (-1, 0)$. The proof can be found in Appendix B.

**Proposition 1 (Descent Lemma)** *Suppose* $\beta \in (-1, 0)$, $I_{Stein}(\rho_n \mid \pi) \geq 2\delta$ *and Assumptions 1, 2, 4 and 5 hold. Choosing*

$$\begin{cases} 0 < \gamma \leq \frac{1}{6(C_{\rho_n}(\delta) + M_{\rho_n}(\delta))BI_{Stein}(\rho_n|\pi)^{\frac{1}{2}}} \\ 0 < \gamma \leq \frac{2(\beta+1)(I_{Stein}(\rho_n|\pi) - \delta)}{B^2 I_{Stein}(\rho_n|\pi)(LM_{\rho_n}(\delta)^2 + 10(C_{\rho_n}(\delta) + M_{\rho_n}(\delta))^2)} \\ 0 < \gamma \leq \frac{\beta+1}{B^2\left(LM_{\rho_n}(\delta)^2 + 10(C_{\rho_n}(\delta) + M_{\rho_n}(\delta))^2\right)} \end{cases}, \tag{20}$$

*we have the descent property*

$$e^{\beta D_{\beta+1}(\rho_{n+1}|\pi)} - e^{\beta D_{\beta+1}(\rho_n|\pi)} \geq -\beta(\beta+1)\gamma \left( \frac{1}{2} I_{Stein}(\rho_n \mid \pi) - \delta \right). \tag{21}$$

Proposition 1 contains the descent lemma for the population limit SVGD Liu (2017); Korba et al. (2020). Actually, let $\beta$ and $\delta$ approach to 0, the descent lemma for the population limit SVGD will be derived by L'Hospital rule. When $\beta > 0$, we also have Equation (21), however due to the sign change of $-\beta$, Equation (21) can not guarantee $D_{\beta+1}(\rho_{n+1} \mid \pi) < D_{\beta+1}(\rho_n \mid \pi)$ anymore (for an asymptotic analysis, please refer to Appendix C).

**Remark 3** *The lack of a descent lemma for $\beta$-SVGD when $\beta > 0$ is not a great loss for us, as explained in Section 3.1, negative $\beta$ is preferable in the implementation of $\beta$-SVGD. One can see from our experiments that $\beta$-SVGD with negative $\beta$ performs much better than the one with positive $\beta$, this verifies our theory in Section 3.1.*

The next corollary is a discrete time version of Theorem 1. Letting $M_{\rho_n}(\varepsilon)$ and $C_{\rho_n}(\varepsilon)$ have consistent upper bound is reasonable since intuitively $\rho_n$ will approach $\pi$, though we can not verify this beforehand.

**Corollary 2** *In Proposition 1, choose $\delta = \varepsilon$ and suppose Assumptions 1, 2, 3, 4 and 5 hold with uniformly bounded $M_{\rho_n}(\varepsilon)$ and $C_{\rho_n}(\varepsilon)$, so that $\gamma$ is uniformly lower bounded. Then we have at most*

$$N = \Omega \left( -\frac{2}{\beta(\beta+1)\varepsilon\gamma} \right) \tag{22}$$

*iterations to achieve* $\min_{i \in \{0,1,\dots,N\}} I_{Stein}(\rho_i \mid \pi) \leq 3\varepsilon$.

**Remark 4** *We may show that $W_2 \left( \frac{1}{N} \sum_{i=1} \delta_{x_i^n}, \rho_n \right) \to 0$ as $N \to \infty$, following Shi et al. (2021) or Korba et al. (2020). However, the existing methods either give a qualitative analysis or provide a exponential bound under more restrictive assumptions, which can not provide useful information in the implementation of $\beta$-SVGD. Interested readers can refer to Korba et al. (2020); Shi et al. (2021), we will not include them in this paper.*

## 5 CONCLUSION

We construct a family of continuous time flows called $\beta$-SVGD flows on the space of probability distributions, when $\beta \in (-1, 0)$, its convergence rate is independent of the initial distribution and the target distribution. Based on $\beta$-SVGD flow, we design a family of weighted SVGD called $\beta$-SVGD. $\beta$-SVGD has the similar computation complexity as SVGD, but achieves faster convergence rate in our analysis and experiments. We introduce $\beta$-SVGD in this work, but there are still lots of questions we do not answer and deserve to explore, like how to tune the parameters to make it more efficient, the performance of $\beta$-SVGD in more complex model and etc.

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
