# OpenReview forum: "Improved Stein Variational Gradient Descent with Importance Weights"
_ICLR.cc/2023/Conference — Submitted to ICLR 2023_

### Official Review · Reviewer_Nor5 · 2022-10-21

**Confidence:** 4
**Correctness:** 4
**Technical Novelty And Significance:** 2
**Empirical Novelty And Significance:** 2
**Recommendation:** 3

**Clarity, Quality, Novelty And Reproducibility:**

The paper is written clearly and is easy to read. I do not find source code but the amount of experiments is also very minimal.


**Strength And Weaknesses:**

## Strengths:
* Applying importance reweighting to the popular SVGD is a meaningful task that is well-motivated.
* The resulting $\beta$-SVGD algorithm is simple and clean.
* Theoretical analyses demonstrate nice convergence properties of $\beta$-SVGD compared to SVGD.


## Weaknesses:
* It is unclear to me what the optimal $\beta$ is and the theories in Section 3 and Section 4 seem to suggest the opposite of each other. In Theorem 2 we take $\beta = 1$ to obtain exponential convergence under the Stein Poincare inequality, while in Proposition 1 we need $\beta \in (-1, 0)$.
* The selling point of the main Theorem 1 is that with $\beta \neq 0$ there is no dependence on $D_{KL}(\rho_0 || \pi)$. However, I do not consider (14) to be a strong result due to the minimization over $t \in [0,T]$ on the left side. As Korba et al. (2020) commented on their Proposition 3 (which is straightforward from their (12)), "the convergence of $I_{Stein}(\mu_t | \pi)$ itself can be arbitrarily slow" unless with more assumptions.
* It is also unclear to me whether the proofs contain novel techniques or are simple adaptations of existing techniques, especially given a large number of assumptions. As mentioned in the paragraph before Remark 1, the importance weights $(\pi / \rho_t)^\beta$ is to cancel exactly the inverse term appearing in the Wasserstein gradient of $(\beta+1)$-Renyi divergence. Are there other insights in the proofs other than this kind of cancellation which seems by design?
* The algorithm is not very practical in my opinion, due to the presence of $\pi/\rho_t$ in the importance weights. In many applications (e.g. Bayesian inference with lots of samples), $\pi$ can be numerically insignificant whereas only $\log \pi$ is numerically reliable. Furthermore, the authors claim we can ignore the normalizing constant in $\pi$, yet this constant can change the step size drastically and must be addressed (this is not a problem of SVGD which only uses the score). Additionally, as already observed by the authors, $\rho_t$ can be too small when using kernel density estimation and can cause overflow --- a rather ad-hoc clipping trick is adopted but I doubt its effectiveness. A more practical reweighting scheme is by Xu et al. [1] and it should be compared to.
* The biggest weakness of the paper, to me, is that the paper has not demonstrated the effectiveness of the algorithm in any applications, despite the simple form of the algorithm. An extremely simple 1-D example is done in Figure 1 and in the appendix, and even then the proposed method does not seem to work well (in Figure 1 the green line is quite off from the target blue line). The authors wrote "more complicated models on high dimension space are out of our interest and ability" in the appendix, but such a statement is irresponsible. At the very least the authors should test on multivariate Gaussians in dimension 2 and above, and compute metrics (energy distance or Wasserstein distance) to evaluate the sample quality.


[1] Xu, L., Korba, A. &amp; Slepcev, D.. (2022). Accurate Quantization of Measures via Interacting Particle-based Optimization. <i>Proceedings of the 39th International Conference on Machine Learning</i>, in <i>Proceedings of Machine Learning Research</i> 162:24576-24595 Available from https://proceedings.mlr.press/v162/xu22d.html.


**Summary Of The Paper:**

This paper proposes enhancing SVGD using importance weights which scales the SVGD update directions by $(\pi / \rho_t)^\beta$. The resulting $\beta$-SVGD is then shown to have an exponential convergence rate in terms of 2-Renyi divergence under a Stein Poincare inequality assumption when $\beta = 1$ and another descent lemma is proved for $\beta \in (-1, 0)$ under quite a few assumptions.


**Summary Of The Review:**

Overall I think the proposed algorithm is well-motivated but its effectiveness has not been demonstrated, both in theory and in practice. Hence I'm leaning toward rejection.

---

> ### Author Response · Authors · 2022-11-06
> **Some misunderstanding of our paper**
>
> Thanks for the reviewer's comment.
>
> 1: section 3 studies the continuous dynamics, this includes the cases $\beta\in (-1,\infty)$. In theorem 1, we prove when $\beta\in (-1,0)$, the time-averaged Stein Fisher information is bounded by $-\frac{1}{T\beta(\beta+1)}$, while the cases $\beta\in [0,\infty)$ depend on $\rho_0$ and $\pi$. For the case $\beta=1$, we prove a exponential convergence rate under Stein-Poincare inequality in theorem 2. Section 4 is devoted to the algorithm, in this section, we only consider $\beta\in (-1,0)$, since the continuous dynamic in section 3 shows the advantage when $\beta\in (-1,0)$.
>
> 2. The criterion is the time-averaged Stein Fisher information~(compare with the nonconvex optimization literature, the criterion there is the time averged $\||\nabla f(x_t)\||^2$) or the minimum of the Stein Fisher information in the interval $[0,T]$. We don't think $I_{Stein}(\rho_T\mid\pi)$ is a proper criterion, since without more assumptions, it is impossible to have accurate information about $I_{Stein}(\rho_T\mid\pi)$.  By Markov inequality, we have at least probability $(1-\delta)$ to have $I_{Stein}(\rho_t\mid\pi)\leq-\frac{1}{T\beta(1+\beta)\delta}$, where $t$ is uniformly sampled from $[0,T]$, while for SVGD, we only have $I_{Stein}(\rho_t\mid\pi)\leq-\frac{KL(\rho_0\mid\pi)}{T\\delta}$ with the same probability, which is a huge improvement.
>
> 3. The assumptions in our paper are messy but to get a concrete guarantee we rely on them and they are hard to remove right now. The main thread is following the proof of Liu[2017] but with some differences. The main difficulty of the proof is that the gradient of Renyi divergence is no longer linear, so we need to do some transformation and use Jensen inequality to get the bound.
>
> 4. We admit that the KDE method may not work well in high dimensions. The Stein importance weight from "Black-Box Importance Sampling" may be a better choice. I don't quite know what do you mean by "A more practical reweighting scheme is by Xu et al. [1]", seems they also use KDE and have the same issue in high dimension.
>
> 5. We agree that the performance of $\beta$-SVGD is not been checked in any real applications, so far the experiments in the paper are used to verify Theorem 1.  Please note in figure 1, in the left picture, $\beta=0.5$, the middle one is $\beta=0$ that is SVGD, and $\beta=-0.5$, figure 1 is used to verify Theorem 1, and the bad performance when $\beta=0.5$ is predicted by Theorem 1. More experiments to support theorem 1 are in the appendix, in all the experiments, we compare $\beta$-SVGD algorithm with $\beta=0.1,0,-0.5$, and $0.5$-SVGD performs the worst while $-0.5$-SVGD performs the best, this supports the Theorem 1.
>
> about the code: we include the link to our code in the last section of the appendix, please check it carefully.

---

### Official Review · Reviewer_xAMu · 2022-10-23

**Confidence:** 5
**Correctness:** 3
**Technical Novelty And Significance:** 2
**Empirical Novelty And Significance:** 2
**Recommendation:** 6

**Clarity, Quality, Novelty And Reproducibility:**

The quality of this paper is in the middle. Under the assumption, many convergence results can be formulated directly.
I think the interesting part is either the analytical or numerical implementation of the methods. More numerical results are excepted.

**Strength And Weaknesses:**

Strength: The preconditioner function (importance weights) is an interesting idea.

Weaknesses:

1. Please do not use the red color in the abstract for the KL divergence.

2. The authors miss much important literature in modified Wasserstein gradient flows. Please discuss them in the literature.

(1) A. Garbuno-Inigo, F. Hoffman, W. Li, A. Stuart, Interacting Langevin Diffusions: Gradient Structure And Ensemble Kalman Sampler, 2019.
(2) W.C. Li, L.X. Ying. Hessian transport gradient flows. Research in the Mathematical Sciences, 2019.



**Summary Of The Paper:**

The paper studies a generalization of Stein variational gradient descent by a particular preconditioner function (weight function). The objective is the same as the KL divergence. By assumptions, they show the convergence result. They also demonstrate the effectiveness of the method in simple numerical examples.

**Summary Of The Review:**

The paper is overall written well with clear mathematics. However, much-related literature and many machine learning-related numerical examples are missing. I strongly suggest authors revise the paper accordingly.

---

> ### Author Response · Authors · 2022-11-06
> **Reference and numerical examples**
>
> Thanks for the reviewer's comments.
>
> The papers you mentioned are quite related, and we will add them in the later version. We also believe that more numerical experiments are needed, we will add them in the later version.

---

> > ### Comment · Reviewer_xAMu · 2022-11-23
> > **Reply to authors**
> >
> > Thanks for addressing my concerns. Numerical examples are needed. I will raise my score from 5 to 6.

---

### Official Review · Reviewer_1o9E · 2022-10-23

**Confidence:** 3
**Correctness:** 2
**Technical Novelty And Significance:** 2
**Empirical Novelty And Significance:** 1
**Recommendation:** 1

**Clarity, Quality, Novelty And Reproducibility:**

The paper is written clearly.

Some notations are confusing.
- Equation 19: $ \left(\frac{\rho_n}{\pi}\right)^\beta(x)\left(\frac{\pi}{\rho_n}\right)^\beta \wedge M_{\rho_n}(\delta) $ should be $\left(\frac{\rho_n}{\pi}\right)^\beta(x) \left[ \left(\frac{\pi}{\rho_n}\right)^\beta \wedge M_{\rho_n}(\delta) \right] $.

- Assumption 5 $\nabla\left( \frac{\pi}{\rho_n} \right)^\beta$: it is unclear whether the power is outside or inside the gradient operator.

**Strength And Weaknesses:**

**Strength**

- The paper proves the convergence of the proposed algorithm.

**Weakness**
- In order to compute the weight $\pi/\rho_t$, the paper proposes to use a kernel density regression to estimate $\rho_t$. However, the number of samples needed by kernel density regression grows exponentially with dimension. So the algorithm won't apply in high dimensions.

- The paper only has toy examples (1-dimensional 2-Gaussian mixture) in the experiments. SVGD is expected to work for much more challenging problems, not for such simple tasks.

- The mathematical assumptions needed for the proof are too strong. For example, for Assumption 4 to hold for some uniform bound M, it essentially requires that $\pi$ and $\rho_0$ cannot be very different.

**Summary Of The Paper:**


The paper proposes and studies an algorithm, $\beta$-SVGD, which modifies the SVGD by multiplying a factor $-(\frac{\pi}{\rho_t})^\beta$ to the vector field. The main result is that when $\beta\in(-1,0)$, the convergence of the Stein Fisher information does not depend on the target $\pi$ or the initialization $\rho_0$.

**Summary Of The Review:**

The paper proposes $\beta$-SVGD but is unable to show that the algorithm works in higher dimensions or more complex problems.
The convergence rate proof depends on unrealistic assumptions.

---

### Author Response · Authors · 2022-11-06
**Concerns on the calculation of the importance weight**

The reviewers were concerned about the practicality of using the KDE method to estimate $\frac{\pi}{\rho_t}$, especially when $d$ is large and the normalization constant of $\pi$ is unknown.  Stein importance weight~(Liu&Lee) $N\omega_i, i=1,\ldots, N$, which is a consistent estimator of $\frac{\pi}{\rho_t}$, could be a good replacement.

The calculation of $\omega_i$ only involves $\nabla\log(\pi)$ and the complexity to get $\omega_i$ is of order $\frac{N^2}{\epsilon}$ by quadratic optimization method~(details can be found in Liu&Lee).

ref: Liu&Lee: "Black-Box Importance Sampling"

---

### Decision · Program_Chairs · 2023-01-20

**Decision:**

Reject

**Justification For Why Not Higher Score:**


The paper can not be accepted because it does not meet the standards of quality, novelty, and significance expected for a machine learning conference. The paper does not provide sufficient evidence or justification for its claims, and does not address the challenges and limitations of its method. The paper also does not show any clear advantage or benefit of its method over existing methods, and does not test its method on realistic or relevant problems. The paper needs major revisions and improvements to be considered for publication.

**Justification For Why Not Lower Score:**

N/A

**Metareview: Summary, Strengths And Weaknesses:**


The paper proposes a variant of Stein variational gradient descent (SVGD) that uses importance weights to scale the update directions. The authors claim that the proposed method, -SVGD, has better convergence properties than SVGD in terms of 2-Renyi divergence and Stein Fisher information. They also provide some theoretical analyses and a simple numerical example to support their claims.

The reviewers appreciate the motivation and simplicity of the proposed method, but they also raise several major concerns that prevent the paper from being accepted. These concerns include:

- The lack of clarity and consistency in the choice of the parameter  and its relation to the convergence results. The paper seems to suggest different values of  for different scenarios, but does not provide any guidance or justification for choosing the optimal .
- The weakness and unrealisticness of the theoretical results. The main theorem relies on a minimization over  that can make the convergence arbitrarily slow, and the other results depend on strong assumptions that are unlikely to hold in practice. The reviewers also question the novelty and insight of the proof techniques, which seem to rely on simple cancellations by design.
- The impracticality and inefficiency of the algorithm. The paper does not address the numerical challenges of computing and normalizing the importance weights, which involve the evaluation of the target density  and its gradient. The paper also uses a crude clipping trick to avoid overflow, but does not justify its effectiveness or robustness. The reviewers suggest comparing the proposed method with a more practical reweighting scheme by Xu et al. (2022).
- The lack of empirical evaluation and demonstration. The paper only provides a very simple 1-D example that does not show any advantage of the proposed method over SVGD. The paper does not test the method on any high-dimensional or complex problems, which are the typical applications of SVGD. The reviewers expect the paper to provide more convincing and comprehensive experiments to validate the method.

In summary, the paper fails to demonstrate the significance and novelty of the proposed method, both theoretically and empirically. The paper also suffers from several technical and practical issues that limit its applicability and usefulness. Therefore, the paper is rejected.